# Unveiling the Correlations between Clinical Assessment of Spasticity and Muscle Strength and Neurophysiological Testing of Muscle Activity in Incomplete Spinal Cord Injury Patients: The Importance of a Comprehensive Evaluation

Katarzyna Leszczyńska [1,2,*,†] and Juliusz Huber [1,†]

1  Department of Pathophysiology of Locomotor Organs, Poznan University of Medical Sciences, 28 Czerwca 1956 no 135/147, 60-545 Poznań, Poland; juliusz.huber@ump.edu.pl
2  Department of Neurosurgery, Wroclaw Medical University, Borowska 213, 50-556 Wroclaw, Poland
*  Correspondence: kat.leszczynska@gmail.com
†  These authors contributed equally to this work.

**Featured Application: Investigating the correlations between clinical assessments and neurophysiological testing for improved rehabilitation outcomes in patients after incomplete spinal cord injury.**

**Abstract:** Spasticity and muscle weakness are prevalent symptoms of incomplete spinal cord injury (iSCI) and can significantly impact patients' quality of life. Clinical spasticity and muscle strength assessments are often used to monitor iSCI patients' progress and plan rehabilitation interventions. However, these assessment methods are subjective, may have limited accuracy, and may not provide a detailed understanding of the underlying neurophysiological changes that occur following spinal trauma. In this study, we aimed to explore correlations between standard clinical assessments of spasticity and muscle strength and objective, non-invasive neurophysiological measures of muscle activity using surface electromyography (sEMG) in iSCI patients up to 2 months after injury. We evaluated 85 iSCI patients (ASIA C = 24, and D = 61) $1.3 \pm 0.3$ months after C3-L1 spinal injury and 80 healthy volunteers (for comparison), using standard clinical assessment tools such as the Modified Ashworth Scale (MAS) and the Lovett Scale (Lovett), and neurophysiological tests, including surface electromyography at rest (rsEMG) and during the attempt of maximal contraction (mcsEMG) performed in chosen key muscles for the trunk (rectus abdominis), upper (abductor pollicis brevis), and lower extremities (rectus femoris and extensor digitorum brevis). We analysed pain in Visual Analog Scale (VAS) and also performed electroneurography to evaluate the peripheral motor impulse transmission. We confirmed a similar level of pain and moderate advancement of axonal injury type in all patients, which, therefore, had no significant effect on the differences in the assessment of patients' muscle activity. Considering evaluation of the iSCI patients in the early post-traumatic stage, depending on the level of the injury, the highest MAS and rsEMG values and the lowest Lovett and mcsEMG scores were found in C3–C5 iSCI patients in most of the key muscles. Patients with Th7–L1 injuries represented moderate MAS and rsEMG results, while the muscle strength and motor units' activity were the worst in the extensor digitorum brevis muscle. Patients with Th3–Th6 incomplete injuries generally presented a moderate level of muscle pathology compared to the above groups. Considering results in all patients, we found strong positive correlations between MAS and rsEMG ($r_{\varepsilon} = 0.752$, $p = 0.009$), and Lovett and mcsEMG ($r_s = 0.602$, $p = 0.008$) results, and negative correlations between rsEMG and mcsEMG scores ($r_s = -0.504$, $p = 0.008$) and MAS and Lovett ($r_s = -0.502$, $p = 0.03$). The changes in muscle motor units' properties, recorded in rsEMG and mcsEMG, although they follow a similar pattern, are, however, different depending on the level of injury in an early post-traumatic stage of iSCI patients. The established correlations between clinical evaluations and neurophysiological assessments, as well as electromyography at rest and during the attempt of maximal contraction, depict a fundamental phenomenon that should be considered during the initial stages of formulating rehabilitation strategies in applied medicine. The

value of neurophysiological sEMG testing seems to be superior to the standard clinical assessment in evaluating spasticity and muscle strength decrease as pathological symptoms found in iSCI patients. Neurophysiological testing, including sEMG, offers a more comprehensive and precise characterisation of muscle activity, thereby enabling the detection of subclinical changes that may otherwise go unnoticed.

**Keywords:** incomplete spinal cord injury; spasticity; electromyography; Modified Ashworth Scale; Lovett Scale

## 1. Introduction

Spasticity and muscle weakness are prevalent symptoms of incomplete spinal cord injury (iSCI) and can significantly impact patients' quality of life [1]. Weakness results from partial paralysis which occurs when certain motor pathways are disrupted while others remain intact, such as in iSCI cases. This form of weakness can have a profound impact on patients' daily life [2]. Spasticity, frequently observed as a concurrent condition in iSCI, is defined by increased muscle tension and spasm that vary based on the speed of movement. These manifestations occur due to uncontrolled reflex activity in muscles located below the level of the injury [3]. The perplexity surrounding spasticity and its associated phenomenon in clinical practice calls for a more practical and more objective approach to address this matter [4]. Clinical assessments of spasticity and muscle strength, such as the Modified Ashworth Scale (MAS) and the Lovett Scale (manual muscle testing, Lovett), are often used to monitor iSCI patients' progress and plan rehabilitation algorithms [5]. However, these methods of assessment are subjective, may have limited accuracy, and may not provide a detailed understanding of the underlying neurophysiological changes that occur after spinal trauma [6,7]. Following spinal cord injury, individuals who are experiencing spasticity and decreased muscle strength require targeted interventions during the rehabilitation period. Treatment approaches typically involve a combination of pharmacological and physiotherapeutic interventions aimed at mitigating the symptoms and improving functional outcomes [8–12]. Some neuromodulation methods also appear to be effective [13–16].

Clinical assessment of iSCI is a part of the American Spinal Injury Association–ASIA scale (ASIA) [17]. However, it is important to have alternative tools to assess the health status of these patients more accurately in order to be able to modify applied therapy on an ongoing basis and tailor it to one's individual needs [18].

The clinical outcomes of iSCI depend on the severity and location of the lesion and may include partial or complete loss of sensory and/or motor function below the level of injury. Lower thoracic lesions can cause paraplegia, while lesions at the cervical level are often associated with semi or total quadriplegia [19]. This study has been undertaken to verify the above symptoms and dysfunctions in an early post-traumatic period. It can be supposed that the present general picture of the iSCI consequences may differ if the development of pathologies is evaluated in an early or late period after the injury.

Most of the pathological symptoms and the spontaneous functional recovery phenomena develop during the first 3 months and, in most cases, reach a plateau by 9 months after injury; additional recovery may also occur up to 12–18 months post-injury [20]. Long-term neurophysiological outcomes of iSCI may be closely related to the level of the injury, the severity of the primary injury, and the progression of secondary injury, and they have been characterised in our previous studies with the usage of sEMG recordings from upper and lower extremities muscles [14–16]. The description of functional pathological consequences in muscle motor units in patients with iSCI immediately after spinal cord injury has been rarely described, which is another of the main reasons for undertaking the sEMG recordings in this work.

Increased muscle tension at rest is commonly associated with the decreased motor units' muscle contractile properties [5]. The precise aetiology of the aforementioned phenomenon remains unclear, as it has not been fully elucidated in both clinical and neurophysiological investigations. Muscle weakness may be the result of spasticity or the consequence of direct injury of the lower motor neuron in iSCI patients. From the physiotherapeutic point of view, the explanation of this relationship may influence the choice of the kinesiotherapy approach—either resulting in a reduction of muscle tension or in the implementation of isometric exercises in paralysed muscles [18,21].

Both the Lovett Scale and MAS have limited precision due to their relatively small number of rating degrees. The Lovett Scale has only six degrees, while MAS has five, which may limit their ability to provide a detailed description of the patient's condition. Many clinical studies have attempted to determine the reliability and consistency of measurements and the credibility of these scales in cases of patients with increased muscle tension, such as after stroke [22], cerebral palsy [23], multiple sclerosis [24], or iSCI [12,25]. Inter- and intra-rater studies provided different results, including differences in the MAS application to the upper and lower extremity muscles [26]. The feasibility and utility of MAS are also discussed with reference to the examination of the trunk muscles [27].

The neurophysiological approach with sEMG recordings to assess both increased muscle tone and decreased contractile properties of the muscles seems to further clarify the relationship between these symptoms. The correlation between the aforementioned characteristics of spasticity and muscle strength has been noted in a prior study in stroke patients [22]. However, it remains unverified whether this hypothesis holds true in patients with iSCI. Moreover, the available literature provided different data on the validity of the sEMG approach in the evaluation of iSCI patients' health status [6,7]. Therefore, this paper attempts to find the relationship between clinical and neurophysiological methods of spasticity evaluation and abnormalities in motor muscle units' activity in patients with iSCI.

## 2. Materials and Methods

### 2.1. Patients and Healthy Subjects

Starting in 2015, we created a database of more than 600 cases of people with incomplete spinal cord injury (iSCI) treated surgically in the same medical centre and diagnosed with clinical and neurophysiological methods by the same investigators. This study describes the results of an evaluation of 85 iSCI patients (American Spinal Injury Association—ASIA scale; ASIA C = 24, and D = 61), $1.3 \pm 0.3$ months after C3–L1 spinal injury (Table 1).

**Table 1.** Demographic, anthropometric, and traumas characteristics of the iSCI patients and healthy volunteers from the control group. Minimum, maximum, mean values and standard deviations are presented.

| Subjects | Age (Years) | Height (cm) | Weight (kg) | Averaged Time from Injury (Months) | ASIA AIS Score | Injury Level |
|---|---|---|---|---|---|---|
| Patients N = 85 ♀= 19, ♂= 66 | 18–64 39.3 ± 9.5 | 153–194 175.7 ± 7.9 | 49–96 61.4 ± 5.3 | 1–2 1.3 ± 0.3 | C = 24 D = 61 | C3–C5 = 16 C6–Th1 = 19 Th3–Th6 = 18 Th7–L1 = 32 |
| Control N = 80 ♀= 17, ♂= 63 | 18–60 38.1 ± 8.4 | 155–189 173.6 ± 6.3 | 45–91 60.9 ± 4.2 | NA | NA | NA |
| p | 0.08 | 0.09 | 0.08 | NA | NA | NA |

Abbreviations: $p < 0.05$ determines significant statistical differences marked bold; NA—non-applicable.

The patients were chosen considering the spinal cord levels, their similar degree of injury based on neuroimaging results (the preservation of the spinal cord structure in

1/3 to 1/4), age, gender, and the average time from trauma. We created four groups of subjects with the levels of injury C3–C5, C6–Th1, Th3–Th6, and Th7–L1. The group division was based on the innervation level of selected key muscles and the location of motor centres relative to neuromeres innervating these muscles. Moreover, this division ensured a balanced distribution and the highest possible iSCI incidence frequency. All patients were studied once with standard clinical assessment tools such as the Modified Ashworth Scale (MAS) and the Lovett Scale (Lovett) and neurophysiological tests, including surface electromyography at rest (rsEMG) and during the attempt of maximal contraction (mcsEMG) performed in chosen key muscles for the trunk (rectus abdominis), upper (abductor pollicis brevis), and lower extremities (rectus femoris and extensor digitorum brevis). Additionally, we analysed pain in Visual Analog Scale (VAS) and also performed electroneurography (ENG) to evaluate the peripheral motor impulse transmission in the median and the peroneal nerves. All patients included in the study were either in the very initial stages of their structured rehabilitation process (scheduling and organising such treatment) or prior to the commencement of such treatment. Patients who had sustained the injury more than two months prior to their enrollment in the study or those who had already begun an intensive rehabilitation program were excluded from the project. The other exclusion criteria were electronic implants and devices such as a pacemaker, an insulin pump, a baclofen pump, or a cochlear implant, stroke, episodes of plexopathies during treatment, inflammatory diseases, confirmed COVID-19 related disorders, and myelopathies diagnosed before or after the incident. None of the patients used medications that could affect the assessment of their spasticity (e.g., spasticity-reducing drugs) or their muscle strength evaluated using both clinical scales and neurophysiological tests. All the patients were informed and understood the potential for the risk of all the procedures. The study was conducted according to the guidelines of the Declaration of Helsinki and was approved by the Bioethics Committee of the Medical University (decision no. 942/2021).

The control group of healthy volunteers (N = 80) was examined to obtain the reference values of control clinical and neurophysiological recordings. The gender, age, height, and weight of the subjects were adjusted for better comparison to the study group. There were no statistically significant differences between the age, height, and weight of patients in the study groups and healthy volunteers in the control group (Table 1).

All patients and healthy subjects understood that there was no financial benefit from participation, and they signed a written consent form for voluntary participation in the study.

### *2.2. Clinical Assessment Tools*
#### 2.2.1. Lovett Scale

The Lovett Scale, often called the manual muscle testing technique, is a tool clinically used to assess muscle strength [28]. It focuses on muscle groups rather than individual muscles. Each muscle group is graded on a 0–5 scale, with 0 indicating no muscle contraction and 5 indicating normal strength. To administer the Lovett Scale for muscle strength assessment, a trained examiner applies manual resistance to each muscle group and grades the strength based on the individual's ability to overcome the resistance. In our study, the test was attempted three times by two investigators, and the best result was agreed as the final score. All muscle groups in the upper and lower extremities with the methodological principles described in the previous studies were tested [27,29]. In cases of the abductor pollicis brevis muscle, it was the abduction of the thumb; for rectus femoris, it was raising the leg in with the knee extended; and for extensor digitorum brevis upper, it was dorsal flexion of the great toe and the ankle. The evaluation of muscle strength of rectus abdominis relied on the visual observation of the patient's ability to bend the torso from the lying position forward without help from the hands [30].

2.2.2. Modified Ashworth Scale (MAS)

The Modified Ashworth Scale (MAS) is a commonly used tool to assess spasticity in individuals. It measures the resistance of the affected muscle group to passive stretch and ranges from 0 to 4 (0, 1, +1, 2, 3, 4), with 0 indicating no increase in muscle tone and 4 indicating rigid, fixed contracture [26,31]. The MAS was administered by the two examiners who passively stretched the affected muscle group and assigned a score based on the level of the resistance felt during the stretch. The common score was agreed upon and included in the analysis.

*2.3. sEMG and ENG Recordings*

This section outlines the methodological principles of bilateral surface electromyography (sEMG) recordings used to evaluate muscle tension at rest (rsEMG) and motor unit recruitment during the attempt of a 5-s maximal contraction (mcsEMG), as depicted in Figure 1. The selection of specific muscles for surface electromyography (sEMG) recordings, considering the level of spinal cord injury, relied on the findings described by Balbinot et al. [32,33]. To conduct the sEMG recordings, we utilised the KeyPoint Diagnostic System (Medtronic A/S, Skøvlunde, Denmark) while patients were in a supine position. These tests took place in a temperature-controlled room maintained at an average of 22 °C. For sEMG measurements, we employed standard disposable Ag/AgCl surface recording electrodes with an active surface area of 5 mm$^2$. The active electrode was positioned on the muscle belly, the reference electrode on the distal tendon of the same muscle, and the ground electrode on the distal part of the examined muscle, following the Guidelines of the International Federation of Clinical Neurophysiology—European Chapter [34,35]. The recording system was configured with upper and lower filters set at 10 kHz and 20 Hz, respectively. The sEMG examination involved two stages. Initially, patients were instructed to lie down comfortably and endeavour to relax the muscles being examined (see Figure 1A(a–d). Subsequently, patients were guided to contract the targeted muscles and, at the examiner's command, exert the maximum possible contraction, sustaining it for 5 s (see Figure 1B(a–d). Each time, three attempts were made, with a 1-min rest period between attempts. The examiner selected the optimal attempt for analysis based on the highest mean amplitude measured peak-to-peak relative to the isoelectric line. The recorded output included the amplitude measured in microvolts (μV) and the frequency of muscle motor unit action potential recruitment measured in Hertz (Hz). A frequency index (ranging from 3 to 0) was assigned according to the calculations of motor unit action potential recruitment during maximal contraction in the sEMG recording: 3 = 95–70 Hz (normal); 2 = 65–40 Hz (moderate abnormality); 1 = 35–10 Hz (severe abnormality); 0 = no contraction. Both control subjects and patients underwent sEMG recordings using a base time of 80 ms/D and an amplification range of 20–1000 μV/D (see Figure 1C–F).

In this particular study, bilateral electroneurography (ENG) was conducted to evaluate the transmission of neural impulses in the peripheral motor fibres of the median and peroneal nerves. The objective was to determine if any notable differences in nerve conduction existed that could potentially impact the assessment of muscle function. The procedure involved administering rectangular pulses lasting 0.2 ms at a frequency of 1 Hz, with an intensity ranging from 0 to 80 mA. Bipolar stimulating electrodes were positioned on the skin along the anatomical pathways of the nerves. Compound muscle action potentials (CMAP potentials) were then recorded from the abductor pollicis brevis (APB) and extensor digitorum brevis (EDB) muscles. To record the evoked potentials, surface electrode pairs were employed, using the same type of electrodes used for the sEMG recordings. The measurements were performed with an amplification range of 5–5000 μV and a time base of 2–10 ms. Normative values obtained from healthy volunteers were subsequently compared with the test results from the patients. The recorded outcomes comprised the amplitudes (in μV) and latencies (in ms) of the M-wave recordings. The electroneurographical assessment of impulse transmission in motor fibres was expressed using a three-grade

score: 0—indicating no transmission, 1—representing severe abnormalities, 2—signifying moderate abnormalities, and 3—denoting normal transmission.

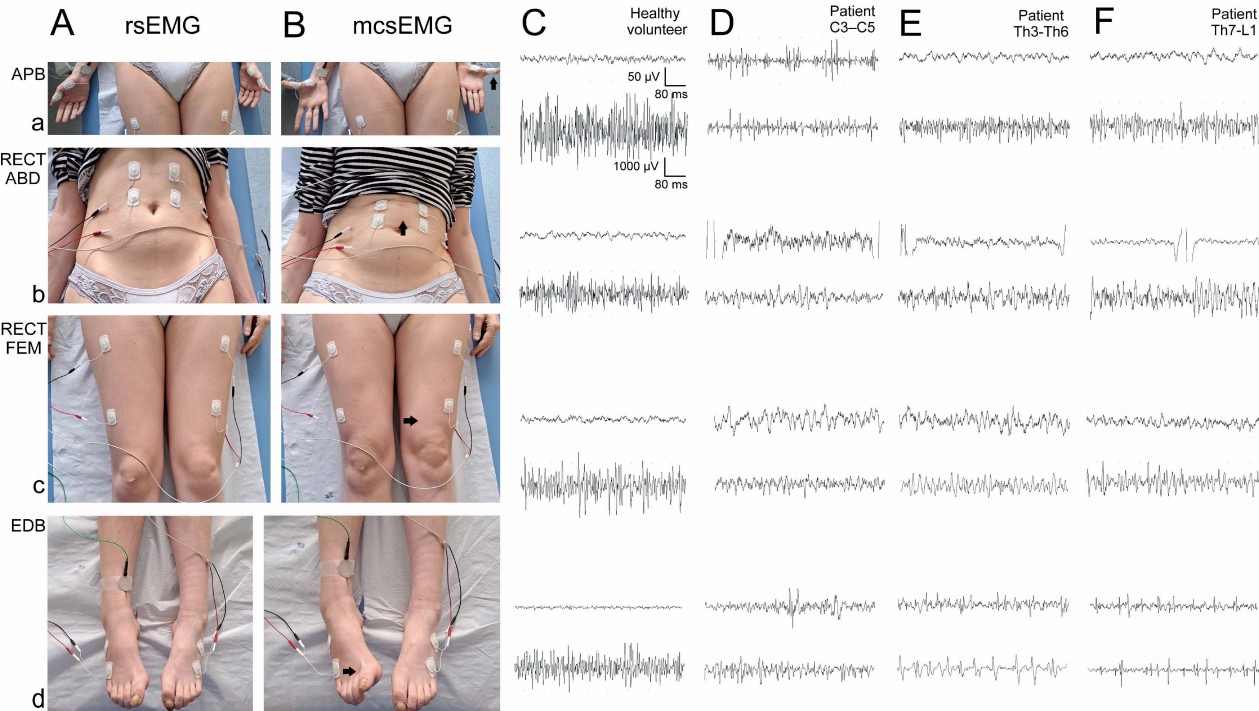

**Figure 1.** The principles of sEMG methodology with pairs of bilateral electrodes placed bilaterally during recordings from a—abductor pollicis brevis (APB), b—rectus abdominis (RECT ABD), c—rectus femoris (RECT FEM), d—extensor digitorum brevis (EDB); (**A**)—sEMG performed at rest (rsEMG), (**B**)—sEMG performed during the attempt of maximal contraction (mcsEMG) in a healthy volunteer (**C**) and in patients with iSCI at different spinal levels (**D**–**F**). Each upper recording in (**C**–**F**) was recorded at rest, while the lower was during the attempt of the maximal contraction. Arrows in (**B**) indicate the signs of the 5-s attempt of maximal contraction. Calibration bars for the amplification and the time base presented in (**C**) are the same for all the recordings.

*2.4. Statistical Analysis*

The data analysis was performed using Statistica, version 13.1 (StatSoft, Kraków, Poland). Descriptive statistics, including mean values, standard deviations (SD), and minimum (min) and maximum (max) values, were calculated for measurable variables. Data mining techniques were employed to match patients and healthy volunteers based on age, sex, weight, and height, as basic anthropomorphic properties. To assess the normality distribution and homogeneity of variances, Shapiro–Wilk tests and Levene's tests were conducted. In patients from four groups with incomplete spinal cord injuries at the cervical and thoracic levels, the mean values of parameters obtained from sEMG tests and clinical tests were compared using Student's $t$-test and the Mann–Whitney test. Statistical significance was determined at $p < 0.05$, indicating significant differences. Preliminary statistical analysis was conducted to determine the required sample size. The primary outcome variables from sEMG recordings of the anterior tibialis muscles were used, with a power of 80% and a significance level of 0.05 (two-tailed). The mean and standard deviation (SD) were calculated based on data from twenty subjects in both patient and healthy control groups. The sample size estimation software indicated a minimum requirement of 70 subjects in each group. To explore the correlations between Ashworth's or Lovett's scale scores and the rsEMG or mcsEMG amplitude measurement results, respectively, the non-parametric Spearman's rank correlation coefficient (rs) was employed. A significance level of $p < 0.05$ was considered statistically significant for rank correlation.

## 3. Results

Overall, patients reported pain on the VAS from 0 to 5 with an average of $1.78 \pm 1.0$ (see Table 2); it was the most increased (an average of 3.0) in C3–C5 patient groups and the least (an average of 0.9) in Th7–L1 group.

**Table 2.** Comparison of the results of pain and electroneurographical assessment in healthy volunteers and iSCI patients. Minimum, maximum, mean values and standard deviations are presented.

| Subjects | Injury Level | VAS Score (0–10) | ENG Score (0–3) |
|---|---|---|---|
| Patients<br>N = 85<br>♀= 19, ♂ = 66 | C3–C5 = 16<br>C6–Th1 = 19<br>Th3–Th6 = 18<br>Th7–L1 = 32 | $0–5\ 3.0 \pm 1.6$<br>$0–3\ 1.9 \pm 0.9$<br>$0–3\ 1.3 \pm 0.9$<br>$0–2\ 0.9 \pm 0.7$ | $1–3\ 2.1 \pm 0.9$<br>$0–3\ 1.5 \pm 0.9$<br>$0–3\ 1.7 \pm 1.0$<br>$0–3\ 1.3 \pm 0.8$ |
| Control<br>N = 80<br>♀= 17, ♂ = 63 | NA | 0–0 0 | 3–3 3.0 |
| *p* | NA | **0.04–0.01** | **0.04–0.03** |

Abbreviations: $p < 0.05$ determines significant statistical differences marked bold; NA—non-applicable; VAS—visual analogue scale (0—no pain, 1–3—mild pain, 4–6—moderate pain, 7–10—severe pain); ENG—electroneurographical evaluation of impulses transmission in motor fibres (0—no transmission; 1—severe abnormalities; 2—moderate abnormalities; 3—normal transmission).

Analysis of the ENG results indicated more moderate than severe scores of the abnormality in the peripheral motor transmission of neural impulses in fibres of upper and lower extremities nerves. Similarly, as in VAS results, they were the most increased at 2.1 on average in patients from the C3–C5 group and the least at 1.3 on average in the Th7–L1 group.

Considering clinical and neurophysiological results recorded in all iSCI patients, we found significant differences for all measured parameters at $p = 0.01$ for the MAS score and at $p = 0.02$ for the Lovett Scale results (Table 3). Neurophysiological studies provided evidence of much more greatest differences, especially for the parameters of rsEMG amplitudes at $p = 0.009$ and mcsEMG amplitudes at $p = 0.008$, indicating the pathological function of muscle motor units in comparison to the healthy subjects. Considering the parameter of mcsEMG of motor units firing frequency (FI), the differences were the most significant (at $p = 0.03$) in recordings from lower extremity muscles, the least from APB muscles.

Analysing the results recorded in certain groups of patients with injuries at different levels of the spinal cord in comparison to the healthy subjects, the highest MAS and rsEMG values, as well as the lowest Lovett Scale and mcsEMG scores, were found in C3–C5 iSCI patients in most of the key muscles. Patients with Th7–L1 injuries represented moderate MAS and rsEMG results, while the muscle strength and motor units' activity appeared to be the worst in the extensor digitorum brevis muscle (see Figure 1F). Patients with Th3–Th6 incomplete injuries generally presented a moderate level of muscle pathology compared to the above groups (Table 4; see an example in Figure 1E). The graphical presentation of the measured parameters' variability is shown in the charts in Figure 2. In general, the results of all parameters from clinical and neurophysiological studies are statistically different from the normative recorded in the control groups, but the least in recordings from APB muscles in patients from the Th3–Th6 and Th7–L1 groups.



**Table 3.** Data on results from clinical and neurophysiological studies recorded in certain groups of muscles of all patients with incomplete spinal cord injuries. Cumulative data from the left and right sides as ranges, means, and standard deviation values are presented. Reference values from results recorded in the control subjects are presented below.

| Examined Muscle | MAS (0-4) | rsEMG Amplitude (µV) | Lovett (0–5) | mcsEMG Amplitude (µV) | FI (3–0) |
|---|---|---|---|---|---|
| **All iSCI Patients (N = 85)** | | | | | |
| APB | 1–4 1.3 ± 0.7 | 5–150 26.5 ± 7.9 | 1–5 4.3 ± 1.3 | 50–5000 1907.2 ± 122.1 | 1–3 2.3 ± 0.8 |
| RECT ABD | 1–3 1.3 ± 0.5 | 5–150 24.9 ± 12.2 | 1–5 2.8 ± 1.4 | 0–2000 466.2 ± 102.3 | 0–3 1.6 ± 0.6 |
| RECT FEM | 1–4 1.5 ± 0.6 | 15–25 31.3 ± 14.9 | 0–5 1.9 ± 1.1 | 0–4000 322.4 ± 151.4 | 0–3 1.4 ± 0.6 |
| EDB | 1–3 1.4 ± 0.5 | 10–25 29.2 ± 14.1 | 0–5 1.4 ± 0.9 | 0–3000 277.4 ± 137.8 | 0–3 1.0 ± 0.7 |
| **Total average** | 1.4 ± 0.6 | 27.9 ± 12.3 | 2.6 ± 1.1 | 743.3 ± 128.4 | 1.5 ± 0.7 |
| **Healthy volunteers—Control (N = 80)** | | | | | |
| **Examined muscle** | **MAS (0–4)** | **rsEMG Amplitude (µV)** | **Lovett (0–5)** | **mcsEMG Amplitude (µV)** | **FI (3–0)** |
| APB | 0–1 0.1 ± 0.3 | 10–25 16.5 ± 1.7 | 5–5 5.0 | 900–6000 2725.3 ± 112.2 | 3–3 3.0 |
| RECT ABD | 0–1 0.4 ± 0.2 | 15–25 17.1 ± 2.0 | 4–5 4.9 ± 0.5 | 400–2100 1098.2 ± 212.1 | 2–3 2.8 ± 0.4 |
| RECT FEM | 0–0 0 | 15–25 17.4 ± 1.3 | 4–5 4.9 ± 0.6 | 800–4200 1725.1 ± 189.2 | 2–3 2.9 ± 0.3 |
| EDB | 0–0 0 | 10–25 16.3 ± 1.8 | 5–5 5.0 | 900–3500 1456.9 ± 191.4 | 3–3 3.0 |
| **Total average** | 0.1 ± 0.2 | 16.8 ± 1.7 | 4.9 ± 0.5 | 1751.3 ± 176.2 | 2.9 ± 0.3 |
| *p* **All iSCI patients vs Control** | **0.01** | **0.009** | **0.02** | **0.008** | **0.03** |

Abbreviations: iSCI-incomplete spinal cord injury; APB—abductor pollicis brevis muscle; RECT ABD-rectus abdominis muscle; RECT FEM-rectus femoris muscle; EDB—extensor digitorum brevis muscle; MAS—evaluation of the muscle tension with the Modified Ashworth Scale (0, 1, +1, 2, 3, 4); rsEMG—sEMG recording at rest; Lovett—evaluation of the muscle's strength with the Lovett Scale (0, 1, 2, 3, 4, 5); mcsEMG—sEMG recording during maximal contraction; FI—frequency index (3–0)—frequency of motor units action potentials recruitment during maximal contraction (3–95–70 Hz—normal; 2–65–40 Hz—moderate abnormality; 1–35–10 Hz—severe abnormality; 0—no contraction); $p < 0.05$ determines significant statistical differences marked bold.

**Table 4.** Comparison of results from clinical and neurophysiological studies in a group of healthy volunteers and in the groups of patients with incomplete spinal cord injuries at different levels. Cumulative data from the left and right sides as ranges, means, and standard deviation values are presented.

| Examined Muscle | Test | Healthy Volunteers (Control) | Patients C3–C5 | Patients C6–Th1 | Patients Th3–Th6 | Patients Th7–L1 | *p* Control vs. Patients C3–C5 | *p* Control vs. Patients C6–Th1 | *p* Control vs. Patients Th3–Th6 | *p* Control vs. Patients Th7–L1 |
|---|---|---|---|---|---|---|---|---|---|---|
| APB | MAS (0–4) | 0–1<br>0.1 ± 0.3 | 1–4<br>2.5 ± 1.3 | 1–4<br>1.5 ± 0.8 | 1–4<br>1.5 ± 0.8 | 1–3<br>1.0 ± 0.2 | **0.01** | **0.03** | **0.03** | **0.04** |
| | rsEMG Amplitude (µV) | 10–25<br>16.5 ± 1.7 | 15–150<br>61.4 ± 14.8 | 10–150<br>36.2 ± 11.8 | 10–100<br>17.3 ± 5.4 | 5–100<br>18.1 ± 8.2 | **0.009** | **0.01** | 0.06 | 0.05 |
| | Lovett (0–5) | 5–5<br>5.0 | 1–5<br>2.4 ± 1.4 | 1–5<br>3.2 ± 1.7 | 2–5<br>4.1 ± 0.7 | 4–5<br>4.8 ± 0.1 | **0.01** | **0.03** | 0.05 | 0.07 |
| | mcsEMG Amplitude (µV) | 900–6000<br>2725.3 ± 112.2 | 50–1000<br>375.0 ± 154.1 | 50–2500<br>831.1 ± 120.3 | 200–4000<br>2320.2 ± 287.1 | 800–5000<br>2120.1 ± 192.7 | **0.008** | **0.009** | **0.04** | **0.04** |
| | FI (3–0) | 3–3<br>3.0 | 1–3<br>1.6 ± 0.8 | 1–3<br>1.7 ± 0.8 | 1–3<br>2.6 ± 0.7 | 1–3<br>2.6 ± 0.6 | **0.01** | **0.01** | 0.05 | 0.05 |
| RECT ABD | MAS (0–4) | 0–1<br>0.4 ± 0.2 | 1–3<br>1.6 ± 0.6 | 1–3<br>1.4 ± 0.5 | 1–3<br>1.4 ± 0.5 | 1–3<br>1.1 ± 0.3 | **0.02** | **0.03** | **0.03** | **0.04** |
| | rsEMG Amplitude (µV) | 15–25<br>17.1 ± 2.0 | 20–35<br>27.4 ± 6.4 | 15–100<br>29.4 ± 9.8 | 10–100<br>25.3 ± 9.4 | 10–60<br>20.5 ± 9.6 | **0.02** | **0.01** | **0.04** | 0.05 |
| | Lovett (0–5) | 4–5<br>4.9 ± 0.5 | 1–4<br>2.1 ± 1.1 | 1–4<br>1.9 ± 1.1 | 1–5<br>2.8 ± 1.3 | 1–5<br>3.6 ± 1.4 | **0.02** | **0.02** | **0.03** | **0.04** |
| | mcsEMG Amplitude (µV) | 400–2100<br>1098.2 ± 212.1 | 50–500<br>216.6 ± 65.3 | 0–600<br>242.5 ± 68.1 | 100–1200<br>422.7 ± 150.1 | 100–2000<br>672.0 ± 76.6 | **0.008** | **0.009** | **0.01** | **0.03** |
| | FI (3–0) | 2–3<br>2.8 ± 0.4 | 1–2<br>1.6 ± 0.5 | 0–3<br>1.5 ± 0.6 | 1–3<br>2.1 ± 0.5 | 1–3<br>2.1 ± 0.7 | **0.01** | **0.01** | **0.03** | **0.03** |
| RECT FEM | MAS (0–4) | 0–0<br>0 | 1–3<br>1.8 ± 0.5 | 1–4<br>1.7 ± 0.6 | 1–4<br>1.7 ± 0.7 | 1–3<br>1.3 ± 0.5 | **0.02** | **0.02** | **0.02** | **0.03** |
| | rsEMG Amplitude (µV) | 15–25<br>17.4 ± 1.3 | 20–45<br>33.4 ± 12.1 | 10–100<br>34.2 ± 14.1 | 15–100<br>31.3 ± 8.9 | 10–60<br>26.8 ± 10.3 | **0.02** | **0.02** | **0.02** | **0.03** |
| | Lovett (0–5) | 4–5<br>4.9 ± 0.6 | 1–4<br>2.1 ± 1.3 | 0–5<br>1.9 ± 1.6 | 0–5<br>1.8 ± 1.3 | 0–5<br>1.8 ± 1.4 | **0.01** | **0.01** | **0.01** | **0.01** |
| | mcsEMG Amplitude (µV) | 800–4200<br>1725.1 ± 189.2 | 50–500<br>219.2 ± 73.8 | 0–1500<br>290.2 ± 80.1 | 0–1500<br>282.1 ± 141.5 | 0–4000<br>390.1 ± 68.2 | **0.009** | **0.01** | **0.01** | **0.02** |
| | FI (3–0) | 2–3<br>2.9 ± 0.3 | 1–2<br>1.5 ± 0.5 | 0–3<br>1.2 ± 0.6 | 0–3<br>1.5 ± 0.6 | 0–3<br>1.4 ± 0.6 | **0.009** | **0.008** | **0.009** | **0.009** |
| EDB | MAS (0–4) | 0–0<br>0 | 1–3<br>1.6 ± 0.7 | 1–3<br>1.3 ± 0.4 | 1–3<br>1.3 ± 0.4 | 1–3<br>1.4 ± 0.6 | **0.02** | **0.03** | **0.03** | **0.03** |
| | rsEMG Amplitude (µV) | 10–25<br>16.3 ± 1.8 | 20–50<br>31.9 ± 14.4 | 15–50<br>27.3 ± 5.1 | 15–60<br>28.0 ± 10.1 | 10–100<br>28.9 ± 14.2 | **0.01** | **0.02** | **0.02** | **0.01** |
| | Lovett (0–5) | 5–5<br>5.0 | 1–5<br>1.9 ± 1.4 | 0–5<br>1.5 ± 1.2 | 0–5<br>1.5 ± 0.6 | 0–5<br>1.1 ± 0.9 | **0.009** | **0.009** | **0.009** | **0.008** |
| | mcsEMG Amplitude (µV) | 900–3500<br>1456.9 ± 191.4 | 0–800<br>245.2 ± 68.1 | 0–1200<br>273.0 ± 48.6 | 0–3000<br>345.2 ± 149.1 | 0–900<br>115.1 ± 61.5 | **0.009** | **0.009** | **0.01** | **0.008** |
| | FI (3–0) | 3–3<br>3.0 | 0–2<br>1.2 ± 0.6 | 0–2<br>1.1 ± 0.5 | 0–3<br>1.1 ± 0.8 | 0–2<br>0.9 ± 0.5 | **0.009** | **0.008** | **0.008** | **0.008** |

Abbreviations: iSCI-incomplete spinal cord injury; APB—abductor pollicis brevis muscle; RECT ABD-rectus abdominis muscle; RECT FEM-rectus femoris muscle; EDB—extensor digitorum brevis muscle; MAS—evaluation of the muscle's tension with Modified Ashworth's Scale (0, 1, +1, 2, 3, 4); rsEMG—sEMG recording at rest; Lovett—evaluation of the muscle's strength with Lovett's scale (0, 1, 2, 3, 4, 5); mcsEMG—sEMG recording during maximal contraction; FI—frequency index (3–0)—frequency of motor units action potentials recruitment during maximal contraction (3–95–70 Hz—normal; 2–65–40 Hz—moderate abnormality; 1–35–10 Hz—severe abnormality; 0—no contraction); $p < 0.05$ determines significant statistical differences marked in bold.

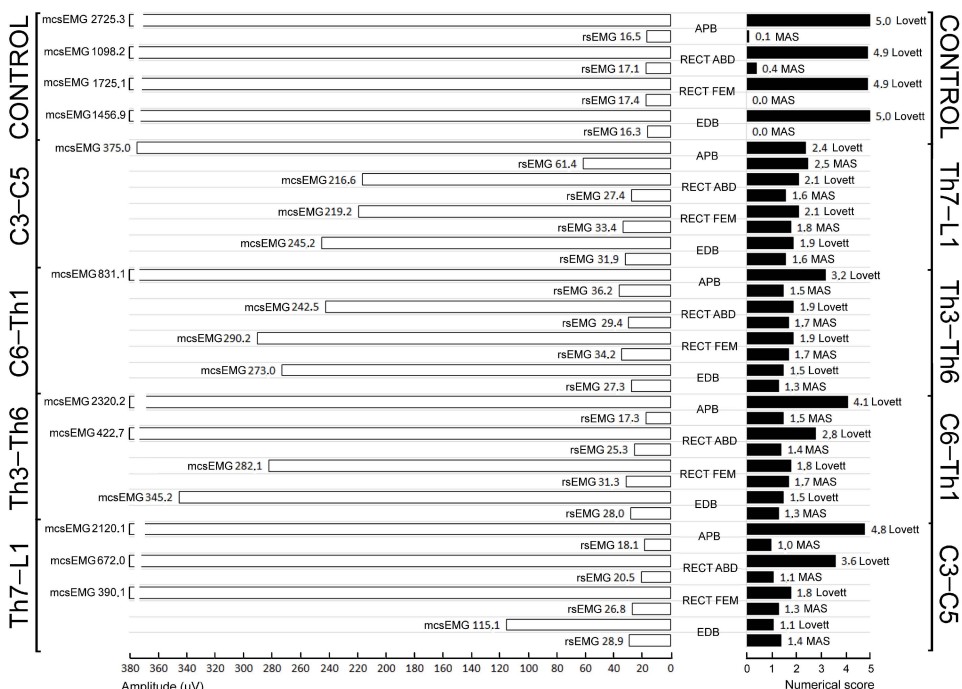

**Figure 2.** Graphical presentation of the mean measured parameters that are summarised in detail in Table 3, with reference to the healthy volunteers (control) and iSCI patients divided into four study groups. Some of the bars describing the values shown on the left side are cut intentionally; the real values are provided in numbers.

The correlation study results of the examined parameters recorded in the clinical and neurophysiological studies are presented in Table 5, while their graphical presentation is in charts of Figure 3A–C. We have found strong positive correlations between MAS and rsEMG (rs = 0.752, $p$ = 0.009), as well as Lovett and mcsEMG (rs = 0.602, $p$ = 0.008) results, and negative correlations between rsEMG and mcsEMG scores (rs = −0.504, $p$ = 0.008). The relation between MAS and Lovett also statistically confirmed (rs = −0.502, $p$ = 0.03) is not fully consequent as the result of neurophysiological correlation.

**Table 5.** Spearman's rank correlation ($r_s$) calculated for the sEMG measurements and clinical study results in the four groups of patients. Cumulative data from the right and left sides are presented. $p \leq 0.05$ was assumed as statistically significant for rank correlation.

| Parameter | MAS | |
|---|---|---|
| **rsEMG** | $r_S$ | $p$ |
| | 0.752 | 0.009 |
| | **Lovett** | |
| **mcsEMG** | $r_S$ | $p$ |
| | 0.602 | 0.008 |
| | **MAS** | |
| **Lovett** | $r_S$ | $p$ |
| | −0.502 | 0.03 |
| | **rsEMG** | |
| **mcsEMG** | $r_S$ | $p$ |
| | −0.504 | 0.008 |

Abbreviations: iSCI-incomplete spinal cord injury; MAS—evaluation of the muscle's tension with Modified Ashworth's Scale (0, 1, +1, 2, 3, 4); rsEMG—amplitude of sEMG recording at rest; Lovett—evaluation of the muscle's strength with Lovett's scale (0, 1, 2, 3, 4, 5); mcsEMG—amplitude sEMG recording during maximal contraction; $p < 0.05$ determines significant statistical differences.

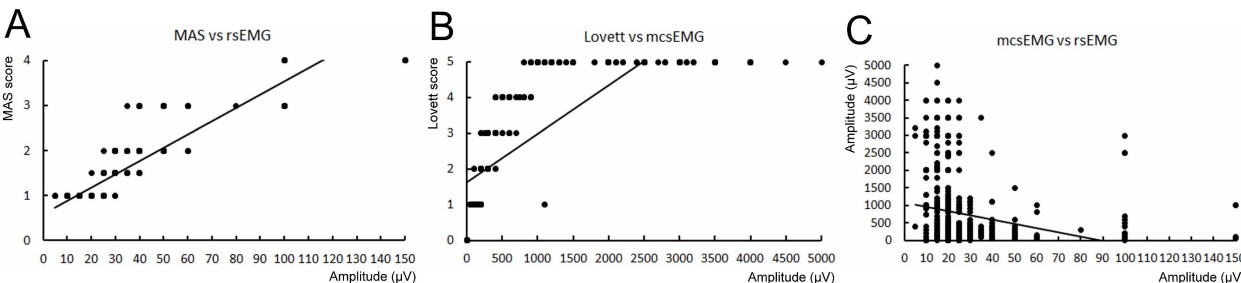

**Figure 3.** Graphical presentation of relationships presented in Table 5; spasticity scores (MAS) vs. resting sEMG amplitude recordings (**A**), muscle strength scores (Lovett) vs. maximal contraction sEMG amplitude recordings (**B**), rsEMG amplitude recordings vs. maximal contraction sEMG amplitude recordings (**C**). Most points on charts are overlapped multiple times with their Y–X locations.

## 4. Discussion

To date, we have not found a publication that addresses all three measures (Lovett Scale, Ashworth Scale, and sEMG) in patients with incomplete spinal cord injury and presents correlations proving the relationship between the various parameters. These relationships, while seeming obvious in clinical work and based on intuition, have so far not had their scientific proof in numbers and data presented based on cases of patients after incomplete spinal cord injuries.

This study supports the notion that a negative relationship exists between increased muscle tension and decreased muscle strength in patients following iSCI with injuries at varying levels, as indicated by standard clinical scale assessments and validated by non-invasive electromyographic testing. If a negative correlation exists between spasticity levels and muscle strength, this might suggest that interventions aimed at reducing spasticity (e.g., pharmacological treatments or stretch therapies) could be beneficial for improving muscle strength. However, it is important to note that correlations do not necessarily indicate causation and that other factors may be influencing the relationship between variables. Therefore, clinical decisions should be based on a comprehensive evaluation of all relevant factors and individual patient characteristics.

The results might strongly depend on two additional pathological factors in addition to the spinal injury itself, namely pain reported by the patients limiting the ability of maximal muscle contraction and changes in transmission of neural impulses within the motor fibres of upper and lower extremity nerves, influencing the number of recruited muscle motor units. The patients in our study reported similar low pain scores on the VAS, with an average of $1.78 \pm 1.0$; the axonal changes in motor fibres were confirmed as moderate in ENG studies (see Table 2). Therefore, both pain and abnormal neural transmission have a very similar impact on muscle tension and contractile muscle properties in all patients analysed in this study. However, it should be remembered that the iSCI patients in our study were just after the injury incidence. The consequences of the spasticity and the pathological, secondary changes in the motor nerve fibres usually occur after three months from the spinal trauma [6,15,36].

Previous studies utilising sEMG and clinical scales for evaluation of the iSCI patients' motor functional status did not provide similar correlations to the ones presented in this study [37,38]. Increased muscle tension is mostly related to the consequences of lower motoneuron dysfunction, which is caused by disturbances in the neural transmission of impulses coming from the supraspinal centres. Therefore, similar observations on the relationship between increased spasticity and decreased muscle strength have been found in patients after ischemic stroke [22]. The lack of the spinal motor centres' control from the excitatory and inhibitory influences of cortico-spinal centres leads to the spontaneous generation of neuronal impulses from the lower motoneurone to the effectors at the rate of 5 Hz [39]. This creates an increase in the amplitude parameter in rsEMG recordings of

more than 25 µV (see Table 3), which neurophysiologically defines the increased muscle tension [40]. Our current study on iSCI patients seems to confirm the above hypothesis.

Analysing the results presented in a chart of Figure 3A and the data in Table 4 referring to the normative values of rsEMG amplitudes recorded in the healthy volunteers for all key muscles under this study, it can be concluded that the amplitude less than 25 µV can be considered as normal, indicating the proper muscle tension. The muscle tension of iSCI patients in our study scored more than 1 in MAS (meaning a slight increase in muscle tone) was not always related to the neurophysiological limit describing the increased muscle tension. None of the iSCI patients in this study received a 0 score in MAS; however, some of them had less than 25 µV-amplitudes in rsEMG recordings, which means normal muscle tensions in terms of neurophysiology. It is important to remember that MAS scale evaluation is affected by the error of subjective assessment. For this reason, objective rsEMG measurements may offer a valuable advantage over MAS in the assessment of patients exhibiting presumed spasticity symptoms due to their superior precision.

Another crucial aspect is that these numerical scales may not be precise enough to detect subtle changes in muscle function after iSCI; it is impossible with the help of Lovett and MAS to analyse subclinical stages. However, in studies by Meseguer–Henarejos et al. [41], the inter- and intra-rater agreement for Modified Ashworth Scale scores were satisfactory. In their studies, MAS scores exhibited better reliability when measuring upper extremities than lower. Several characteristics of the studies were statistically associated with an inter-rater reliability of the scores for the lower and upper extremities. On the other hand, there are inconsistent opinions on the utility of MAS and the Lovett Scale in the clinical evaluation of patients with different movement disorders [42,43]. Akpinar et al. stated that MAS has adequate reliability for determining lower-extremity spasticity in patients with iSCI [43]. On the other hand, Fleuren et al. [44], utilising sEMG for evaluation of the spasticity syndrome, suppose that the validity and reliability of MAS are insufficient to be used as a measure of spasticity. Patients who are evaluated on the Lovett Scale, or MAS, by physiotherapists very often, despite intensive rehabilitation, receive the same score, which does not illustrate real patient progress. Thanks to clinical neurophysiology studies, we are able to evaluate changes in the amplitude and nature of the recording that are subclinical. This may have a significant impact on patients' motivation for further rehabilitation. In addition, it is a tool that gives those in charge of rehabilitation detailed information, thus, a detailed understanding of the nature of the changes and which direction the rehabilitation program should go. In addition, the results obtained in the sEMG test are easier to compare with each other among other researchers, and even the smallest differences, not to be captured in the MAS and Lovett Scales, can be measured. Such an approach highlights the importance of a comprehensive evaluation that may become a key motivation for a patient's further rehabilitation process.

There is little consistent data on the utilisation of sEMG for the evaluation of motor unit activity from different muscle groups of iSCI patients [36,37,45]. The exceptions are the research of Balbinot et al. [32,33], who utilised sEMG to program the algorithm of the diagnostic evaluation of the iSCI patients with different levels of injury, with the rules of recordings interpretations very similar to the ones proposed in this study. Only the study of Latash et al. [38] is comparable with our final results in the evaluation of mcsEMG frequency index (FI) in iSCI patients, pointing at the simultaneous decrease in this parameter together with the amplitude depending on the tested lower extremity muscle groups. However, it is difficult to directly confirm the results of our study with the other previously described utilising sEMG methods.

In cases of iSCI patients, the actual motor function and spinal cord neural transmission recovery is possible due to different mechanisms. They may be the proliferation and sprouting of new axons, which in the early stage of the pathology is limited due to oedema and inflammatory barriers, the conduction of saved axons of the corticospinal tract, or the system of intersegmental cervico–lumbar propriospinal system with crossed projections of long axons at middle-low thoracic levels [46]. We may suppose that in our early injured iSCI

patients at the low thoracic and lumbar levels, in which the motor function is moderately but promisingly expressed in rectus femoris muscles, the propriospinal neurons compensate the abnormalities in the transmission of white matter fibres that finally balance the activity at the motoneuronal level and, consequently, in muscles [46,47].

The primary limitation of this study was the diversity of injuries observed in patients with iSCI, despite our efforts to recruit subjects according to strict enrollment criteria, including evaluation using the ASIA scale and neuroimaging. This is a problem for many research projects that examine the cases of people after spinal cord injuries. Another of the study limitations may also be the possibility of different neurophysiological advancements of post-traumatic pathologies in iSCI patients under this study despite the similarities in structural neuroimaging results. However, it is an unequivocal risk coming from the initial clinical evaluation of patients according to the ASIA scale. Moreover, the recordings of rsEMG and mcsEMG depended on the patient's motivation, but in our study, we attempted the same set of tests three times for each of the muscles in each patient; the final best result was agreed upon by two experienced neurophysiologists. The same held true for the regime of the methodology in cases of clinical evaluation with the MAS and the Lovett Scale. Lastly, neurophysiological testing may be subject to artefacts. Based on our experience and rigorous methodology, we put effort into minimising the occurrence of artifacts during data collection. These measures included proper electrode placement, signal filtering, or quality control procedures based on the Guidelines of the International Federation of Clinical Neurophysiology—European Chapter. By implementing these measures, we aimed to obtain reliable and accurate neurophysiological data. However, we recognise that artefacts can still arise despite our best efforts.

## 5. Summary of Recommendation

We suggest that in addition to clinical evaluation, patients should be examined with neurophysiological tests, including surface electromyography, with the methodology and the algorithm presented in this study. This examination is repeatable, objective, painless, and non-invasive. It provides reliable data that can not only be a source of motivation for the patient but also valuable information for medical professionals involved in iSCI care. Neurophysiology testing offers a more comprehensive and sensitive characterisation of muscle activity than the clinical scores performed alone, thereby enabling the detection of subclinical alterations that may otherwise go unnoticed. We recommend applying the neurophysiological testing in the early post-traumatic stage and repeating similar evaluations in the follow-ups. Such an approach may help to tailor the evidence-based rehabilitation algorithm individually adjusted presently favoured in applied science.

## 6. Conclusions

The changes in muscle motor units' properties, recorded in rsEMG and mcsEMG, are clearly visible depending on the level of injury in an early post-traumatic stage of iSCI patients. The established correlations between clinical evaluations and neurophysiological assessments, as well as electromyography at rest and during the attempt of maximal contraction, depict a fundamental phenomenon that ought to be taken into account during the initial stages of formulating rehabilitation strategies. While working to reduce spasticity, one can also expect to see improvements in muscle strength and function, and vice versa, which has been proven in this study by a strong positive correlation. These findings should be widely applied in clinical practice. Additionally, the value of neurophysiological sEMG testing seems to be superior to the standard clinical assessment in the evaluation of spasticity and muscle strength decrease as pathological symptoms found in iSCI patients. Clinical scales can be a support in the assessment of the patient, but they are subjective, may have limited accuracy, and may not provide a detailed understanding of the underlying neurophysiological changes that occur following spinal trauma. Clinical neurophysiology tests are painless, non-invasive, and easily repeatable, and they enable a more accurate assessment of changes, including subclinical ones.

**Author Contributions:** Conceptualisation, K.L. and J.H.; methodology, K.L. and J.H.; software, J.H.; validation, K.L. and J.H.; formal analysis, K.L. and J.H; investigation, K.L. and J.H; resources, J.H.; data curation, K.L. and J.H.; writing—original draft preparation, K.L.; writing—review and editing, K.L. and J.H; visualization, J.H.; supervision, J.H.; project administration, K.L. All authors have read and agreed to the published version of the manuscript.

**Funding:** This research received no external funding.

**Institutional Review Board Statement:** The study was conducted in accordance with the Declaration of Helsinki and approved by the Bioethics Committee of Poznan University of Medical Science, decision no 942/21 dated 13 January 2022.

**Informed Consent Statement:** Informed consent was obtained from all subjects involved in the study.

**Data Availability Statement:** All the data generated or analysed during this study are included in this published article.

**Conflicts of Interest:** The authors declare no conflict of interest.

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
