# Peer review of "Unveiling the Correlations between Clinical Assessment of Spasticity and Muscle Strength and Neurophysiological Testing of Muscle Activity in Incomplete Spinal Cord Injury Patients: The Importance of a Comprehensive Evaluation"

_applsci, doi:10.3390/app13137609_

Round 1
Reviewer 1 Report
Dear Authors,
Congratulations to the performed work and for the presented algorithm.
Please if is possible, in introduction add more studies about spasticity and muscle strength.
Author Response
Dear Reviewer 1,
Thank you very much for such kind words and positive review. According to your suggestions and comment, we have added more studies about spasticity and muscle strength in the Introduction section (marked in yellow, lines 53-61, and 505-511). Please also consider the in the text marked in colours, which include responses to the suggestions of the other Reviewers.
Kind regards,
Authors
Reviewer 2 Report
Neurophysiological parameters such as sEMG are investigated followed by statistical analysis of muscle activity characterization that primarily benefits clinical practices. there are some minor grammatical errors that need to be rectified. This is an interesting work and I am in favor of its publication.
Minor edits are required.
Author Response
Dear Reviewer 2,
Thank you very much for your kind words about our work and for your positive review. According to your suggestions and comment, we have revised the entire text again and changed the grammar and vocabulary in several places. Please also consider the in the text marked in colours, which include responses to the suggestions of the other Reviewers.
Kind regards,
Authors

Reviewer 3 Report
The authors aim to assess and compare various methods for measuring spasticity and strength with objective techniques (EMG/ENG) in patients who have sustained incomplete spinal cord injury. They conduct a retrospective analysis of patients with incomplete spinal cord injury. Specifically, in their protocol, the select 85 patients from their database that are assessed 2 months post injury. 85 patients were included in this study with ASIA C (24) and D (61) incomplete spinal cord injuries.
VAS and electroencephalography were used to measure peripheral motor impulse transmission, and based on their findings, the authors establish various correlations between the objective measurements (EMG) and subjective scores (Lovett, Ashworth) . They conclude that:
-
During the early stages of rehabilitation, the correlations between clinical evaluations and neurophysiological assessments should be considered to facilitate planning a treatment course for rehab. as well as resting versus maximal contraction EMGs
-
The value of neurophysiological sEMG testing seems to be superior to the standard clinical assessment in evaluating spasticity and muscle strength decrease as patho-logical symptoms found in iSCI patients.
-
Neurophysiological testing offers a more comprehensive and precise characterization of muscle activity, thereby enabling the detection of sub-clinical changes.
Overall, interesting study with well-defined objectives that attempt to establish correlations between the more subjective tests for spasticity and strength and neurophysiological findings.
The study design lends itself to selection bias, as they pre-select 85 patients based on some radiographic criteria (should be better defined) of a database containing more than 600 spinal cord injury patients. A blinded prospective study with larger sample size would allow for more meaningful conclusions.
Comments:
-
Incomplete spinal cord injury can be graded using the ASIA scale from A to D. In this study, there were no ASIA B patients, with the majority being ASIA C and D. The studies as listed are based on correlations, with a small sample pool of patients. While these correlations are apparent at the 2 months time point, it falls short of evaluating the trend over a longer period of time. This raises the question as to the relevance of their findings in the context of how it alters the treatment course from a rehabilitative perspective, and whether the gained knowledge and treatment alterations translate into improved patient outcomes.
-
How do their findings compare to patients at 3 or 6 months post-injury?
-
Neurophysiological parameters are subject to artifact. This should be discussed in the paper and noted as far as potentially affecting the results of the study. The authors do mention pain as being a confounding factor, and this is important given the prevalence of neuropathic pain in such SCI patients. The authors should highlight how other confounders such as medications (e.g used for spasticity) might impact the relevance of their findings, and document if any of the patients assessed were on such medications.
Altogether interesting study but it is limited on the basis of the above concerns. Would not favor publication in its current format for the aforementioned reasons.
Satisfactory.
Reviewer 4 Report
This is a very interesting study with high level of clinical application. I recommend strongly its publication.
Author Response
Dear Reviewer 4,
Thank you very much for your kind words about our work and for your positive review. Please also consider the in the text marked in colours, which include responses to the suggestions of the other Reviewers.
Kind regards,
Authors
Round 2
Reviewer 3 Report
Thank you for addressing the concerns as expressed previously. I support publication of the manuscript in this journal.